# Association between Changes in Protein Intake and Risk of Cognitive Impairment: A Prospective Cohort Study

**DOI:** 10.3390/nu15010002

**Published:** 2022-12-20

**Authors:** Xinyi Xu, Yueheng Yin, Li Niu, Xinxin Yang, Xinru Du, Qingbao Tian

**Affiliations:** 1Postdoctoral Research Station in Basic Medicine, Hebei Medical University, Shijiazhuang 050017, China; 2School of Nursing, Nanjing Medical University, Nanjing 210029, China; 3International Education College, Hebei Medical University, Shijiazhuang 050017, China; 4School of Nursing, Hebei Medical University, Shijiazhuang 050017, China; 5Postdoctoral Research Station in Biology, Hebei Medical University, Shijiazhuang 050017, China; 6Department of Epidemiology and Statistics, School of Public Health, Hebei Medical University, Shijiazhuang 050017, China

**Keywords:** animal-based protein, plant-based protein, change, cognitive impairment, older adults

## Abstract

Little is known about the role of change in protein intake in affecting cognitive function among older adults. Therefore, we aimed to investigate the associations between the change in protein intake from various food groups and cognitive impairment among older adults in a prospective cohort study. A total of 6951 participants without cognitive impairment or dementia were included in this study. The frequency of protein intake from various food groups was measured by a food frequency questionnaire at baseline and follow-up. Multivariable Cox hazard models with time as the underlying time metric applied to calculate the hazard ratios (HRs) and 95% confidence intervals (CIs). During the 37,535 person-years of follow-up, 1202 (17.3%) participants developed cognitive impairment. The improvement in overall protein intake was negatively associated with cognitive impairment with multivariable-adjusted HR of 0.98 (95% CI = 0.97–0.99). Compared with participants with stable change, those with an extreme decline in animal-based protein intake had a 48% higher risk of cognitive impairment. The associations of changes in protein from six food groups with cognitive impairment were in a similar direction to the main result. Protective associations between improving protein intake and a reduced risk of cognitive impairment were observed.

## 1. Introduction

Cognitive impairment is an increasingly significant public health issue. The World Health Organization (WHO) has estimated that the number of people with dementia was 50 million in 2020, and the prevalence will double every 20 years [1]. In 2050, there will be 152 million individuals living with dementia worldwide [2]. Dementia will bring a huge burden to individuals, their families, and health and social care systems [3]. Cognitive impairment has a significant impact on depression, falls, disability, hospitalization, and death among older adults [4]. This situation has attracted considerable attention from the World Health Organization (WHO), which stated that to preserve autonomy and avoid the development of chronic degenerative diseases among older adults, maintaining their normal cognitive function should be prioritized [5].

The Lancet commission concluded that 40% of worldwide dementias can be prevented or delayed by lifestyle factors, including diet and nutrition [3]. Recent systematic reviews suggest that high adherence to some specific diet patterns such as the Mediterranean diet, the Dietary Approach to Stop Hypertension (DASH) diet, and an anti-inflammatory diet might improve cognitive function [6]. However, it is necessary to have a deeper investigation of the components of diet instead of the whole combination, since the consumption of macronutrients can influence cognitive function [7]. Protein is a critical nutrient for normal cognitive functioning [8]. Regarding dietary protein intake, several studies have investigated its association with cognitive function, and the conclusion has been inconsistent. Some studies concluded that protein intake was positively associated with cognitive function [9], whereas some reported null results [7]. Therefore, there is an increasing need to investigate the association between protein intake and cognitive function.

The evidence to support the role of specific protein food groups on cognitive function is still limited [10]. Animal-based and plant-based protein intake has been shown to have diverse associations with well-known risk factors of cognitive impairment, such as hypertension, cardiovascular disease, diabetes, and obesity [11,12]. This implies cognitive impairment may have various associations with protein intake from different food sources. Given the inconsistency and paucity of data, we aimed to investigate the role of protein intake from different food sources on cognitive function in older adults.

Most prospective cohort studies only applied baseline measurements to predict the risk of cognitive impairment at follow-up [9], ignoring dynamic characteristics of protein intake over time, which could potentially introduce measurement errors. From a public health perspective, capturing changes in protein intake is critical because they reflect the risks associated with individuals making lifestyle changes [13,14]. No study to date has been conducted to investigate the association between change in protein intake and cognitive decline. Therefore, the present study aimed to examine the association between change in protein intake from different food groups and cognitive impairment in the older population using the Chinese Longitudinal Healthy Longevity Survey (CLHLS) database.

## 2. Materials and Methods

### 2.1. Study Design, Participants and Procedures

Participants were selected from older adults enrolled in the population-based cohort study titled the CLHLS. The CLHLS was a nationwide prospective cohort study that enrolled individuals aged 65 and older. The sample of CLHLS was randomly selected from 806 cities and counties in 23 provinces of China by using multi-stage stratified sampling, covering about half of the cities and counties in each province [15]. Follow-up surveys were conducted every 3 or 4 years. More detailed information on study design and data quality assessment of the CLHLS has been presented in previous studies [16]. All baseline and follow-up surveys were conducted through face-to-face interviews.

In this study, we included participants from the CLHLS who had normal cognitive function at baseline. The baseline exclusion criteria were people with clinically diagnosed dementia, those with cognitive impairment, missing data regarding the cognitive test, relocation, or death during the follow-up period.

Since the information of egg and nuts intake were first objectively measured in the fifth wave (2008–09), participants in the 2008 to 2009, 2011 to 2012, and 2014 waves, were enrolled in this study. Among 19,419 participants enrolled in the CLHLS from 2008 to 2014, 7074 were excluded since they had dementia or cognitive impairment or had no complete cognitive tests. After we deleted 5394 that died or were lost in follow-up cases, our sample consisted of 6951 participants with normal cognitive function (Figure 1).

### 2.2. Measurement of Protein Intake

The simplified food frequency questionnaire (FFQ) was measured by asking: “How often do you currently consume this food?” The reproducibility and validity of the Chinese food frequency questionnaire have been described previously [17]. Trained personnel were responsible for collecting information on protein food groups that are commonly consumed in China. We divided the protein groups into two categories, including animal-based protein food groups (eggs, fish and aquatic products, meats, and milk and dairy products), and plant-based protein food groups (bean products, nuts) [18]. Food groups were measured by five options, including “almost every day”, “not every day, but at least once per week”, “not every week, but at least once per month”, “not every month, but occasionally”, or “rarely or never”, and the recorded questionnaires were scored between 5 and 1. We computed the animal-based protein intake and plant-based protein intake by summing up food groups accordingly and respectively. Follow-up scores minus baseline scores were identified as changes in protein intake.

The absolute change scores of protein intake were calculated using protein intake at baseline and the first follow-up. According to the distribution of change scores among participants, change patterns included extreme decline (<15th percentile), moderate decline (15–30th percentile), mild decline (30–45th percentile), stable (45–55th percentile), mild improvement (55–70th percentile), moderate improvement (70–85th percentile), and extreme improvement (>85th percentile).

### 2.3. Cognitive Assessment

Cognitive impairment was measured by the Chinese version of the Mini-Mental State Examination (MMSE), adapted and validated from the scale developed by Folstein and colleagues [19]. The Chinese MMSE is reliable and valid for measuring cognitive function among older Chinese adults [20,21], and the validity and reliability of CMMSE were measured and verified in each wave of CLHLS. The reliability of the MMSE scale is high (Cronbach’s a = 0.96) [22]. The Chinese version of MMSE took into account the cultural and socioeconomic status of older adults in China, so that all the questions in the test could be easily understood and answered by survey participants with normal cognitive function [20]. All questions had to be answered by surveyed participants. The CMMSE has made was modified based on the socio-cultural differences of the Chinese population [21,23]. In particular, previous research has shown that participants are more likely to be unable to answer relatively difficult tasks when they exhibit poor health and/or existing cognitive limitations [24]. Therefore, based on previous research, we categorized “unable to answer” responses as incorrect answers. This approach has been widely used in previous studies and did not introduce potential bias [25]. CMMSE measured five aspects of cognitive function (orientation, reaction, attention & calculation, recall, and language) by 24 items. The total score ranged from 0 to 30, and a higher score indicated better cognitive function. Since participants’ average years of schooling in this study was 2.8 ± 5.0 years, we used education-based MMSE cut-off points to screen cognitive impairment, which has been widely used in older adults with low educational levels [26]. The cut-off points of CMMSE defined cognitive impairment were <18, respondents with no formal schooling, <21, respondents with 1 to 6 years of schooling, and <25, respondents with more than six years of schooling [27].

### 2.4. Covariates

Demographic variables, chronic medical conditions, and physical performance are associated with cognitive function in older adults. All multivariate models included the following covariates [28]: age at enrollment, sex, educational level (years of education), residence (urban, rural), socioeconomic status (favorable, unfavorable), marital status (married, divorced/widowed/never), living pattern (living with family members, alone or at nursing home), current smoking behavior (yes, no), current alcohol use (yes, no), current regular physical exercise condition (yes, no), activity of daily living (ADL), the instrumental activity of daily living (IADL), body mass index (BMI), and chronic medical illness including hypertension, diabetes, heart disease, stroke or cardiovascular disease (CVD), cataract, digestive system diseases, arthritis, and Parkinson’s disease.

ADL was measured at each wave using six items (dressing, bathing, indoor transferring, toileting, continence, and feeding). Participants were asked if they needed assistance with each of the six activities. The Katz Index of Independence was applied to assess ADL Disabled, respondents who needed assistance in performing one of the ADLs were considered as ADL disabled [29]. IADL was composed of eight items (shopping, visiting neighbors, washing clothes, making food, walking 1 km, crouching and standing (repeated three times), carrying 5 kg weight, and taking public transport) [30]. According to the Lawton scale, respondents were categorized as having an IADL disability if they needed help performing at least one of the eight items. Items were rated on a three-point scale ranging from 1 (complete independence) to 3 (complete dependence). The higher scores respondents obtained, the greater functional dependence they would have, and would need more external care from the family members or nursing staff.

We calculated BMI as the weight in kilograms (kg) divided by the square of the height in meters (m^2^), categorized into underweight (BMI < 18.5 kg/m^2^), normal (18.5 ≤ BMI < 24 kg/m^2^), overweight (BMI ≥ 24 kg/m^2^) [31].

### 2.5. Statistical Analysis

Descriptive statistics including the Pearson chi-squared test and Student’s t-test were used to summarize the baseline characteristics. “Person-years” were calculated from the time of the baseline survey of participants to the earliest of the following events: the first occurrence of cognitive impairment, death; lost to follow-up, or time of the last survey. We applied Cox hazard models with time as the underlying time metric to calculate the hazard ratios (HRs) and 95% confidence intervals (CIs) for analyzing the association between changes in protein intake (continuous and categorical) and cognitive impairment.

Demographic variables, functional ability, and chronic medical illness were listed as possible covariates. The association between changes in protein intake and cognitive impairment was investigated in three models. Model 1 adjusted for sex and age, Model 2, further adjusted for residence, years of schooling, marital status, economic status, living pattern based on Model 1, and Model 3, further adjusted for smoking, alcohol drinking, ADL, IADL, BMI, and chronic disease (hypertension, diabetes, heart disease, stroke or CVD, cataract, digestive system diseases, arthritis, and Parkinson’s disease) based on Model 2. Adjusted hazard ratios for reversion and 95% confidence intervals were calculated. In addition, we also considered changes in time-varying variables (marital status, economic status, living pattern, smoking, alcohol drinking, ADL, IADL, BMI, and chronic disease) in the Cox hazard models.

Possible non-linear relationships by non-parametrically restricted cubic splines were analyzed between the continuous change points of protein intake and cognitive impairment [32,33]. Four knots were placed at the 15th, 30th, 70th, and 85th percentiles, and we used 0 (no change) as a reference point to test the potential non-linear association of the change in protein intake with cognitive impairment.

We also performed stratified analyses to evaluate potential effect modifications by baseline age (younger elderly at 65–79 years, octogenarian at 80–89 years, nonagenarian and centenarian at ≥90 years), sex (male or female), residence (urban, rural), socioeconomic status (favorable, unfavorable), living pattern (living with family members, alone or at nursing home), current regular physical exercise condition (yes, no), IADL disability (yes, no), and BMI (underweight, normal weight, overweight). We assessed the potential effect modifications by creating a cross-product of the stratifying variable with changes in protein intake in the fully adjusted model.

Analyses were performed with R 4.2.1 (R Foundation for Statistical Computing, Vienna, Austria) and IBM SPSS v26.0 (IBM, Armonk, NY, USA). Statistical tests were two-sided, and *p* values of less than 0.05 were considered to indicate statistical significance.

## 3. Results

Table 1 shows the participant baseline characteristics for those grouped according to the presence or absence of conversion to cognitive impairment at follow-up. During the 37,535 person-years of follow-up, 1202 (17.3%) participants developed cognitive impairment. There was a mean age of 79.7 ± 10.3 years old at baseline, and males accounted for 49.7% of total participants. In total, 2520 (36.3%) participants were urban residents, 3554 (51.2%) were married, 5889 (84.9%) had favorable economic status, and 5665 (84.9%) participants lived with family members. Most demographic variables showed significant differences between converter and non-converters. There were significant differences in change in protein intake (total, animal, and plant) between the two groups with *p* values of <0.001, 0.001, and 0.005, respectively.

As shown in Figure 2, after multivariable adjustment, the non-linearity association between change in overall protein intake and cognitive impairment was insignificant (*p* = 0.122). As shown in Table 2, after multivariable adjustment, the HR of cognitive impairment was 0.98 (95% CI = 0.97–0.99, *p* = 0.001). The associations between change patterns of protein intake and cognitive impairment have been shown in Appendix A.

Change in animal-based protein intake was non-linearly correlated to the risk of cognitive impairment, with an S-shaped relationship (*p* for non-linear trend = 0.019) (Figure 3). After multivariable adjustment, the HR of cognitive impairment was 0.98 (95% CI = 0.97–0.99, *p* = 0.005) (Table 3). Compared with participants with stable change, those who in an extreme decline in animal-based protein intake had a 48% higher risk of cognitive impairment with HR of 1.48 (95% CI = 1.15–1.91, *p* = 0.002), and there was a non-statistically significant increase in risk for cognitive impairment in other groups (*p* > 0.05) (Appendix A). 

In the animal-based protein group, only the change in fish and aquatic products was non-linearly correlated to the risk of cognitive impairment, with a U-shaped relationship (*p* for non-linear trend = 0.018). Compared with participants with stable change, only those in an extreme decline in fish and aquatic products intake had a 50% higher risk of cognitive impairment with HR of 1.50 (95% CI = 1.16–1.93, *p* = 0.002).

There was no significant association between changes in eggs and milk intake and risk of cognitive impairment (*p* > 0.050).

Model 1 was adjusted for age (continuous), and gender, Model 2 was adjusted for model 1 plus residence, years of schooling, marital status, economic status, and living pattern, and Model 3 was adjusted for model 2 tobacco smoking, alcohol drinking, regular exercise, ADL, IADL, and chronic disease (hypertension, diabetes, heart disease, stroke or CVD, cataract, digestive system disease, arthritis, Parkinson’s disease).

The improvement in plant-based protein intake was negatively associated with the risk of cognitive impairment with an HR of 0.96 (95% CI = 0.93–0.99, *p* = 0.010). The non-linearity association between the change in plant-based protein intake and cognitive impairment was insignificant (*p* = 0.902) (Figure 4).

In the plant-based protein group, both changes in bean products and nut intake were non-linearly correlated to the risk of cognitive impairment, with a U-shaped relationship (*p* for non-linear trend = 0.006), and a reverse U-shaped relationship (*p* for non-linear trend = 0.004), respectively. For the change in bean products intake (Appendix A), compared with participants with stable intake, participants with extreme and moderate decline intake had a higher risk of cognitive impairment with HRs of 1.37 (95% CI = 1.09–1.72, *p* = 0.006), and 1.26 (95% CI = 1.01–1.58, *p* = 0.038), respectively. For the change in nut intake, compared with participants with stable intake, participants with mild decline, mild improvement, and moderate improvement intake had a lower risk of cognitive impairment with HRs of 0.81 (95% CI = 0.67–0.97, *p* = 0.025), 0.70 (95% CI = 0.56–0.87, *p* = 0.001), and 0.58 (95% CI = 0.44–0.77, *p* < 0.001), respectively.

After adjusting for changes in time-varying variables, all the associations were similar to the main results (Appendix A). Improvements in total, animal-based, and plant-based protein intake were all negatively associated with the risk of cognitive impairment with HRs of 0.95 (95% CI = 0.92–0.99, *p* = 0.008), 0.98 (95% CI = 0.96–0.997, *p* = 0.021), and 0.96 (95% CI = 0.93–0.99, *p* = 0.007). 

In the subgroup analyses of change in overall protein intake (Table 4), the HRs of cognitive impairment were 0.97 (95% CI = 0.95–0.99, *p* = 0.001) in the octogenarian, and 0.98 (95% CI = 0.97–0.99, *p* = 0.001) in participants living with family members. The negative associations were also significant in people who did not do exercise and who were IADL disabled with HRs of 0.97 (95% CI = 0.96–0.99, *p* < 0.001), and 0.98 (95% CI = 0.97–0.99, *p* = 0.005), respectively. The association between the change in overall protein intake and cognitive impairment was only significant in people in the underweight group with HR of 0.97 (95% CI = 0.95–0.99, *p* = 0.005).

In the subgroup analyses for change in animal-based protein intake, the negative associations were significant in males, urban residents, participants with favorable economic status, participants lived with family members, participants who did not do regular exercise, IADL disabled participants, and underweight participants.

In the subgroup analyses for change in plant-based protein intake, the negative associations were significant in rural residents, participants with unfavorable economic status, participants lived with family members, participants who did not do regular exercise, and IADL abled participants.

## 4. Discussion

In this population-based cohort study, we found that an increase of protein intake was negatively associated with the presence of cognitive impairment after adjusting potential confounders. Meanwhile, an extreme decline in protein intake for most food groups significantly increased the risk of cognitive impairment in subsequent years. This study also shows that changes in animal and plant-based protein intake might have a different impact on different groups of older adults. 

In terms of overall protein, more protein consumption was negatively associated with cognitive impairment. Our findings are consistent with some previous studies. Li et al. (2020) found a positive association between dietary protein intake with cognitive function in adults aged 60 years or older [9]. Glenn et al. (2019) reported that protein intake could maximize the ability to maintain physical activity, and therefore be beneficial for cognitive function [34]. A Harvard study followed more than 77,000 men and women for 20 years, and compared with consuming carbohydrates, eating protein was associated with lower odds of developing cognitive decline later in life [35]. Proteins are the building blocks for muscles, and inadequate protein intake might increase the risk of frailty and sarcopenia, which are closely related to cognitive impairment [36].

In addition, an extreme decline in protein intake could significantly increase the risk of developing cognitive impairment, and a mild or moderate decline and improvement in protein intake were not significantly associated with cognitive impairment in most food groups (meats, fish and aquatic products, eggs, and bean products). To the best of our knowledge, no study has investigated the association between an extreme decline in protein intake and cognitive function in older adults. Even though older adults usually have an age-associated reduction in food intake, their demand for protein increases with age [37,38]. Older adults need more dietary protein to counteract inflammation and catabolism associated with chronic and acute diseases that often occur with aging, and they have a declining anabolic response to protein intake [39]. Therefore, improvement in protein intake might only maintain the current level of cognitive function among older adults during a five-year follow-up. To maintain normal cognitive function with aging, older adults should consume more protein than before instead of keeping the same consumption level. However, it should be noted that currently, regarding to cognitive function, there is no specific recommendation for protein intake for older adults [34]. Additional research is needed to develop definite conclusions of protein intake for maintaining optimal cognitive function in older adults.

Our study also suggested that plant-based protein has a prior impact on cognitive function than animal-based protein, since the HR of plant-based protein intake for lowering the risk of cognitive impairment was also lower than animal-based protein intake, and this result was in line with previous studies [35,40]. Meanwhile, among various protein food groups, only an increase in nut intake decreased the risk of cognitive impairment among older adults. Unlike protein from “red” meats, plant-based protein is not associated with adverse neural consequences due to low-grade systemic inflammation, and therefore was associated with better global cognition in older adults [41]. Tryptophan is an essential amino acid that plays a key role in the microbiota-gut-brain axis, and its metabolites support the development of the central and enteric nervous systems [42]. Tryptophan must be obtained through animal or plant-based protein sources. Some evidence suggested that tryptophan from animal sources appears less readily absorbed by synthetic neurotransmitters than those from plant sources, due to stronger competition with other amino acids [43]. Additionally, the change in milk and dairy products intake was not significantly associated with cognitive impairment in this study. Available evidence on the associations between dairy food consumption and cognitive performance is scarce and inconclusive [44]. Supplementation en Vitamines et Mineraux Antioxidants (SU.VI.MAX) cohorts revealed that total dairy product intake was not associated with cognitive function, and milk consumption was negatively associated with verbal memory performance [45]. By contrast, the Maine-Syracuse Longitudinal Study showed that older adults who consumed the highest amounts of dairy products had better global cognition, executive function, and visuospatial memory compared with those who rarely consumed dairy products [46]. Further investigation of the effects of dairy product intake on cognitive function is required.

Our subgroup analysis showed that the change in protein intake was only effective in older males. Previous studies have shown that regarding physical function, older males benefited more on increasing protein intake than older females [47]. Ogata et al. (2015) found that the association between dairy product intake and short-term memory was only significant among males after adjusting for genetic and family environmental factors [48]. Previous studies summarized that females require a higher baseline starting point protein intake (~1.6 g/kg/day) than males (~1.2 g/kg/day) due to increased protein oxidation [49,50,51]. However, the underlying mechanism of gender difference in the relationship between the change in protein intake and cognitive function was still unclear [52]. Certainly, more studies need to be conducted to ascertain what gender differences in protein metabolism exist and how these differences result in different cognitive outcome.

Another interesting point in this study is that the impact of animal and plant-based protein intake varied by socioeconomic status. The negative associations between the change in animal-based protein intake and cognitive impairment were only significant in older adults with favorable economic status or living in an urban area. In contrast, the negative associations between the change in animal-based protein intake and cognitive impairment were only significant in older adults with unfavorable economic status or living in a rural area. Socioeconomic status such as household income, might play a role in older adults’ dietary preferences and choices of food quality [53]. Consumption frequencies for plant-based protein were significantly associated with lower socioeconomic status in Malaysia and Indonesia [54]. Seafood, meats and dairy products were mostly consumed by the rich [55]. The findings suggest clinical professionals should consider a socio-economic-stratified intervention while promoting protein intake in Chinese older adults.

The improvement in protein intake was not significantly protective for older adults’ cognitive impairment unless they were IADL disabled, and did not do physical exercise. Physical exercise is considered to be the most effective method for maintaining a healthy mind [56]. We hypothesized that if an older adult kept doing regular physical exercise, the impact of a change in protein intake would not be that obvious on cognitive function. In addition, as for the IADL disabled, our findings are consistent with earlier evidence showing that older people who are IADL-disabled have a higher risk of developing cognitive impairment [57], and maintaining high protein intake at an early age is important. These findings suggest that improving protein intake, especially among older adults who are IADL-disabled and without regular exercise, should be viewed as a public health intervention to address cognitive benefits.

To our knowledge, this is the first longitudinal study to examine the association between changes in protein intake from different food groups with the risk of cognitive impairment. However, there still exists some imitations. First, data on cognitive status was self-reported; therefore, it is possible that false-positive results of cognitive impairment and normal cognitive function existed in baseline and follow-ups [58]. Studies that apply objective measurements are needed in the future. Second, the collected dietary information from FFQ lacks quantitative information, which precluded assess to detailed quantitative dietary intake of protein and measurement of macronutrients. This made it impossible to adjust for energy intake in the analyses. However, several key energy intake determinants were considered, such as age, sex, physical activity, ADL, IADL and BMI [59,60]. In addition, the food groups covered the most common sources of dietary protein intake among Chinese older adults [61]. Nevertheless, more specific dietary protein intake information is required in future investigations. Third, cases of death and subjects being lost to follow-up before the first follow-up were deleted, which suggests that these cases were not random and may bias the results. Finally, because the current research design was based on the results of the survey at two-time points, it is unclear whether protein intake maintained cognitive status or whether cognitive status affected protein intake. We did a sensitivity analysis, which directly modeled the changes in protein intake between baseline and the first follow-up and cognitive status in the second follow-up. Only a change in fish and aquatic products intake was significantly negatively associated with cognitive impairment (Appendix A). Since follow-up surveys were conducted every 3 or 4 years in CLHLS, it is necessary to increase the frequency of the time points to examine the relationship between cognitive status and change in protein intake in future research.

## 5. Conclusions

In conclusion, among Chinese older adults, we observed a negative association between improvement in protein intake and risk of cognitive impairment, and extreme decline in protein intake increased the risk of cognitive impairment. Unlike other studies, our investigation highlights the role of improvement and decline in protein intake on cognitive performance in older adults. In addition, the impact of protein intake from different food groups on cognitive function may be affected by the characteristics of older adults. Clinical trials modifying significant protein intake should be conducted to improve the cognitive functions of older adults.

## Figures and Tables

**Figure 1 nutrients-15-00002-f001:**
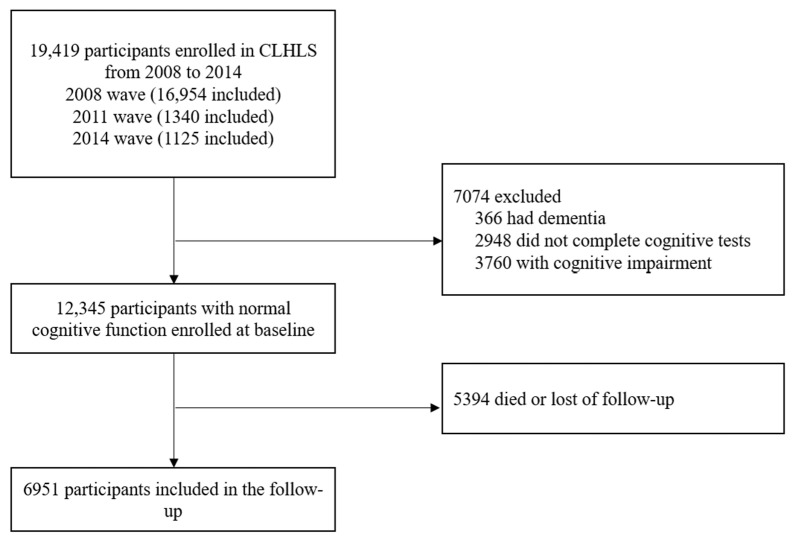
Flow diagram of sample selection.

**Figure 2 nutrients-15-00002-f002:**
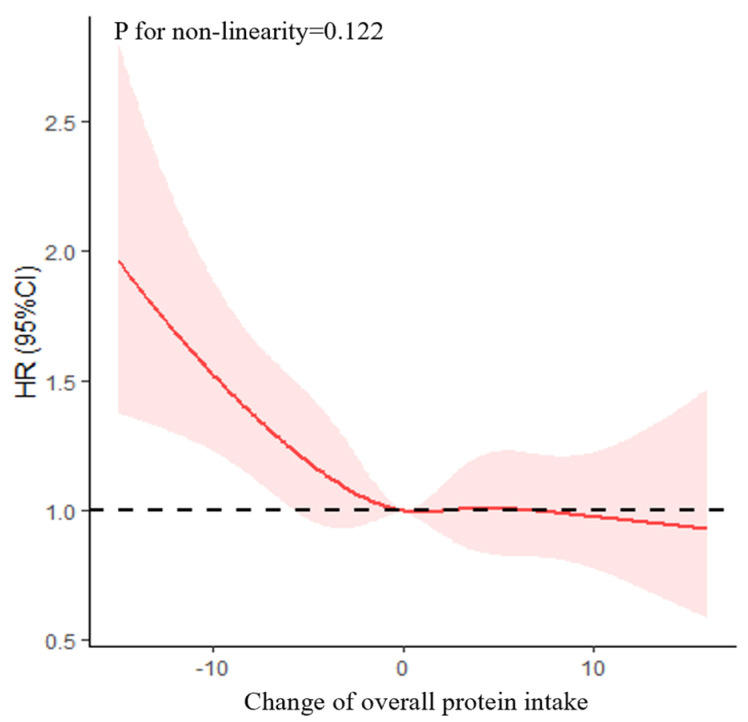
Association between change in overall protein intake and cognitive impairment based on a restricted cubic spline model after adjusting for age (continuous), gender, residence, years of schooling, marital status, economic status, living pattern, tobacco smoking, alcohol drinking, regular exercise, ADL, IADL, BMI, and chronic disease (hypertension, diabetes, heart disease, stroke or CVD, cataract, digestive system disease, arthritis, Parkinson’s disease). The red line represents the HR, the shade of pink represents the 95% CI, and the dotted line represents the reference HR of one.

**Figure 3 nutrients-15-00002-f003:**
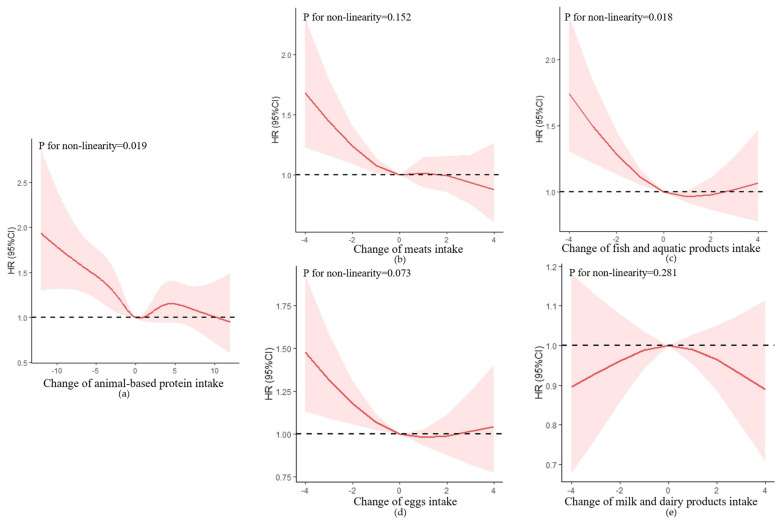
Association between changes in protein intake and cognitive impairment based on restricted cubic spline model after adjusting for age (continuous), gender, residence, years of schooling, marital status, economic status, living pattern, tobacco smoking, alcohol drinking, regular exercise, ADL, IADL, BMI, and chronic disease (hypertension, diabetes, heart disease, stroke or CVD, cataract, digestive system disease, arthritis, Parkinson’s disease). (**a**) Change in animal-based protein intake; (**b**) change in meats intake; (**c**) change in fish and aquatic products intake; (**d**) change in eggs intake; (**e**) change in milk and dairy products intake. The red lines represent the HRs, the shades of pink represent the 95% CIs, and the dotted lines represent the reference HRs of one.

**Figure 4 nutrients-15-00002-f004:**
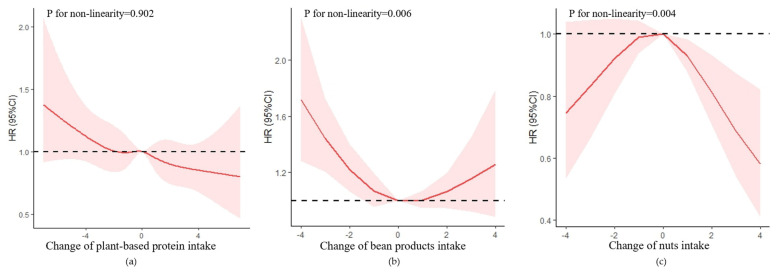
Association between changes in protein intake and cognitive impairment based on restricted cubic spline model after adjusted for age (continuous), gender, residence, years of schooling, marital status, economic status, living pattern, tobacco smoking, alcohol drinking, regular exercise, ADL, IADL, BMI, and chronic disease (hypertension, diabetes, heart disease, stroke or CVD, cataract, digestive system disease, arthritis, Parkinson’s disease). (**a**) Change in plant-based protein intake; (**b**) change in bean products intake; (**c**) change in nuts intake. The red lines represent the HRs, the shades of pink represent the 95% CIs, and the dotted lines represent the reference HRs of one.

**Table 1 nutrients-15-00002-t001:** Baseline characteristics of older people according to cognitive impairment status.

	Total	Cognitive Impairment	Normal	*p* Value
Number of participants	6951	1202	5749	
Age in years	79.7 ± 10.3	86.7 ± 9.4	78.2 ± 9.8	<0.001 ***
Participants per age group				<0.001 ***
Younger elderly	3521 (50.7)	270 (22.5)	3251 (56.5)	
Octogenarian	2092 (30.1)	444 (36.9)	1648 (28.7)	
Nonagenarian and centenarian	1338 (19.2)	488 (40.6)	850 (14.8)	
Male	3456 (49.7)	410 (34.1)	3046 (53.0)	<0.001 ***
Years of schooling	2.8 ± 5.0	1.5 (5.0)	3.1 ± 5.0	<0.001 ***
Urban residence	2520 (36.3)	391 (32.5)	2129 (37.0)	0.003 **
Marital status				<0.001 ***
Married	3554 (51.2)	341 (28.4)	3213 (56.0)	
Divorced/widowed/never	3387 (48.8)		2527 (44.0)	
Economic status				0.024 *
Favorable	5889 (84.9)	992 (82.7)	4897 (85.3)	
Unfavorable	1048 (15.1)	207 (17.3)	841 (14.7)	
Living pattern				<0.001 ***
Living with family members	5665 (81.6)	930 (77.4)	4735 (82.5)	
Alone or at nursing home	1276 (18.4)	272 (22.6)	1004 (17.5)	
ADL	6.1 ± 0.6	6.2 (1.0)	6.1 ± 0.5	<0.001 ***
ADL disabled	303 (4.4)	100 (8.3)	203 (3.5)	<0.001 ***
IADL	10.1 ± 3.6	11.8 ± 4.5	13.1 ± 5.0	<0.001 ***
IADL disabled	2947 (42.4)	786 (65.4)	2161 (37.6)	<0.001 ***
Smoke at present	1582 (22.8)	178 (14.8)	1404 (24.4)	<0.001 ***
Drink alcohol at present	1451 (20.9)	191 (15.9)	1260 (21.9)	<0.001 ***
Exercise at present	2343 (33.8)	317 (26.4)	2026 (35.4)	<0.001 ***
BMI (kg/m^2^)	20.9 ± 3.5	20.1 ± 3.6	21.1 ± 3.4	<0.001 ***
BMI group				<0.001 ***
Underweight	1600 (23.0)	377 (31.4)	1223 (21.3)	
Normal	3952 (56.9)	670 (55.8)	3282 (57.1)	
Overweight	1393 (20.1)	153 (12.8)	1240 (21.6)	
Chronic disease				
Hypertension	1633 (24.0)	273 (23.3)	1360 (24.1)	0.573
Diabetes	208 (3.0)	19 (1.6)	189 (3.3)	0.001 **
Heart disease	637 (9.3)	85 (7.2)	552 (9.8)	0.007 **
Stroke or CVD	352 (5.1)	51 (4.3)	301 (5.3)	0.191
Cataract	520 (7.6)	111 (9.4)	409 (7.2)	0.011 *
Digestive system diseases	336 (5.3)	46 (4.2)	290 (5.6)	0.075
Arthritis	1440 (21.0)	259 (21.9)	1181 (20.8)	0.410
Parkinson’s disease	22 (0.3)	6 (0.5)	16 (0.3)	0.252

* *p* < 0.05 ** *p* < 0.01 *** *p* < 0.001.

**Table 2 nutrients-15-00002-t002:** The association between all variables and cognitive impairment.

Variables	HR (95% CI)	*p* Value
Age in years	1.07 (1.06–1.09)	<0.001 ***
Sex		0.030 *
Female	[1]	
Male	0.84 (0.71–0.98)	
Years of schooling	0.93 (0.90–0.95)	<0.001 ***
Location of residence		0.545
Rural	[1]	
Urban	0.96 (0.84–1.10)	
Marital status		0.001 **
Divorced/widowed/never	[1]	
Married	0.75 (0.63–0.89)	
Economic status		0.348
Unfavorable	[1]	
Favorable	0.92 (0.78–1.09)	
Living pattern		0.545
Alone or at nursing home	[1]	
Living with family members	1.04 (0.89–1.23)	
ADL	1.03 (0.96–1.11)	0.402
IADL	1.03 (1.02–1.05)	<0.001 ***
BMI	1.001 (0.998–1.004)	0.638
Smoke at present		0.859
No	[1]	
Yes	1.02 (0.84–1.10)	
Drink at present		0.482
No	[1]	
Yes	0.48 (0.78–1.12)	
Exercise at present		0.053
No	[1]	
Yes	0.87 (0.75–1.002)	
Hypertension		0.270
No	[1]	
Yes	0.92 (0.78–1.07)	
Diabetes		0.701
No	[1]	
Yes	0.91 (0.56–1.48)	
Heart disease		0.701
No	[1]	
Yes	0.95 (0.74–1.22)	
Stroke or CVD		0.535
No	[1]	
Yes	1.11 (0.80–1.28)	
Cataract		0.899
No	[1]	
Yes	1.02 (0.81–1.28)	
Digestive system diseases		0.375
No	[1]	
Yes	0.87 (0.63–1.19)	
Arthritis		0.305
No	[1]	
Yes	1.08 (0.93–1.26)	
Parkinson’s disease		0.799
No	[1]	
Yes	1.11 (0.49–2.51)	
Change in overall protein intake	0.98 (0.97–0.99)	0.001 **

* *p* < 0.05 ** *p* < 0.01 *** *p* < 0.001.

**Table 3 nutrients-15-00002-t003:** The association between the change in different types of protein intake and cognitive impairment.

	HR (95% CI)	*p* Value
Animal-based protein		
Model 1	0.98 (0.97–0.99)	0.005 **
Model 2	0.98 (0.96–0.99)	0.002 **
Model 3	0.98 (0.96–0.99)	0.003 **
*Meats*		
Model 1	0.95 (0.91–0.995)	0.030 *
Model 2	0.95 (0.91–0.99)	0.018 *
Model 3	0.95 (0.91–0.995)	0.030 *
*Fish and aquatic products*		
Model 1	0.95 (0.91–0.98)	0.005 **
Model 2	0.94 (0.91–0.98)	0.003 **
Model 3	0.94 (0.90–0.98)	0.005 **
*Eggs*		
Model 1	0.96 (0.92–1.001)	0.054
Model 2	0.97 (0.93–1.01)	0.093
Model 3	0.96 (0.92–1.001)	0.052
*Milk* and *dairy products*		
Model 1	0.997 (0.96–1.03)	0.843
Model 2	0.99 (0.96–1.03)	0.728
Model 3	0.99 (0.95–1.03)	0.592
Plant-based protein		
Model 1	0.96 (0.94–0.99)	0.007 **
Model 2	0.96 (0.94–0.99)	0.008 **
Model 3	0.96 (0.93–0.99)	0.010 *
*Bean products*		
Model 1	0.96 (0.93–0.995)	0.023 *
Model 2	0.96 (0.93–0.99)	0.020 *
Model 3	0.96 (0.92–0.996)	0.031 *
Nuts		
Model 1	0.97 (0.93–1.01)	0.096
Model 2	0.97 (0.93–1.01)	0.106
Model 3	0.98 (0.93–1.02)	0.250

* *p* < 0.05 ** *p* < 0.01.

**Table 4 nutrients-15-00002-t004:** The association between change in protein intake and cognitive impairment in subgroups.

			Overall Protein		Animal-Based Protein		Plant-Based Protein	
	No of Converters/Person Years	Conversion Rate	HR (95% CI)	*p* Value	HR (95% CI)	*p* Value	HR (95% CI)	*p* Value
Age								
Younger elderly	270/22,534	7.7	0.98 (0.96–1.01)	0.135	0.98 (0.95–1.02)	0.280	0.96 (0.91–1.01)	0.128
Octogenarian	444/10,083	21.2	0.97 (0.95–0.99)	0.001 **	0.96 (0.94–0.98)	0.001 **	0.96 (0.92–1.004)	0.073
Nonagenarian and centenarian	488/5004	36.5	0.99 (0.97–1.01)	0.225	0.99 (0.96–1.01)	0.291	0.98 (0.93–1.02)	0.323
Sex								
Male	410/18,938	11.9	0.97 (0.95–0.99)	0.005 **	0.96 (0.94–0.99)	0.009 **	0.96 (0.91–1.003)	0.064
Female	792/18,698	22.7	0.99 (0.97–1.001)	0.052	0.98 (0.97–1.002)	0.076	0.98 (0.95–1.01)	0.172
Location of residence								
Urban	391/13,582	15.5	0.98 (0.96–1.003)	0.095	0.98 (0.96–0.99)	0.011 *	0.98 (0.94–1.03)	0.480
Rural	811/24,060	18.3	0.98 (0.97–0.99)	0.003 **	0.98 (0.95–1.002)	0.076	0.96 (0.93–0.99)	0.021 *
Economic status								
Favorable	992/31,977	16.8	0.98 (0.97–0.99)	0.001 **	0.98 (0.96–0.99)	0.005 **	1.02 (0.95–1.09)	0.619
Unfavorable	207/5575	19.8	0.99 (0.96–1.02)	0.543	0.98 (0.94–1.02)	0.281	0.96 (0.93–0.99)	0.010 *
Living pattern								
Living with family members	930/31,157	16.4	0.98 (0.97–0.99)	0.001 **	0.97 (0.96–0.99)	0.003 **	0.97 (0.94–0.99)	0.019 *
Alone or at nursing home	272/6443	21.3	0.99 (0.96–1.02)	0.489	0.99 (0.95–1.02)	0.495	0.99 (0.93–1.05)	0.699
Exercise at present								
Yes	317/13,284	13.5	0.999 (0.98–1.02)	0.962	0.997 (0.97–1.03)	0.825	1.01 (0.96–1.06)	0.792
No	883/24,289	19.3	0.97 (0.96–0.99)	<0.001 ***	0.97 (0.95–0.99)	0.001 **	0.95 (0.92–0.99)	0.004 **
IADL disabled								
Yes	786/13,379	26.7	0.98 (0.97–0.99)	0.005 **	0.98 (0.96–0.995)	0.014 *	0.96 (0.93–0.996)	0.031 *
No	416/24,258	10.4	0.98 (0.96–1.001)	0.061	0.98 (0.95–1.003)	0.078	0.97 (0.93–1.02)	0.230
BMI								
Underweight	377/8144	23.6	0.97 (0.95–0.99)	0.005 **	0.96 (0.94–0.99)	0.005 **	0.96 (0.91–1.01)	0.147
Normal	670/21,617	17.0	0.99 (0.97–1.003)	0.127	0.99 (0.97–1.01)	0.243	0.98 (0.94–1.01)	0.156
Overweight	153/7870	11.0	0.98 (0.94–1.01)	0.149	0.96 (0.92–1.01)	0.114	0.98 (0.91–1.05)	0.524

Converters: participants who converted from normal cognitive function to cognitive impairment. * *p* < 0.05 ** *p* < 0.01 *** *p* < 0.001. Adjusted for age (continuous), gender, residence, years of schooling, marital status, economic status, living pattern, tobacco smoking, alcohol drinking, regular exercise, ADL, IADL, BMI, chronic disease (hypertension, diabetes, heart disease, stroke or CVD, cataract, digestive system disease, arthritis, Parkinson’s disease).

## Data Availability

The datasets analyzed for this study can be found in The Chinese Longitudinal Healthy Longevity Survey (CLHLS)-Longitudinal Data repository, https://opendata.pku.edu.cn/dataset.xhtml?persistentId=doi:10.18170/DVN/WBO7LK (accessed on 28 November 2022).

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
