# Peer review of "Association between Changes in Protein Intake and Risk of Cognitive Impairment: A Prospective Cohort Study"

_nutrients, 2022, doi:10.3390/nu15010002_

Round 1
Reviewer 1 Report
This study was performed to investigate the association between the change in the consumption of protein-rich food and cognitive impairment over time in older adults. As indicated by the authors, available evidence shows that protein intake is associated with cognitive functioning although most of the measurements were taken in a cross-sectional context. In this regard, the potential novelty of this study is related to the examination of “changes” in protein intake from different food groups over time in relation to those of the cognitive function in an older population. The results revealed protective associations between improving protein intake and reducing the risk of cognitive impairment.
Comments
The main concern of this reviewer pertains to the limitations described by the authors about the measurements of the main variables tested in this study. Indeed, in lines 345-351, they highlight the limitation of self-reported cognitive status, and they also emphasize the lack of quantitative information provided by the simplified food frequency questionnaire (FFQ). This obviously raises the question as to “what level of confidence should be given to the relationship between two groups of variables whose the measurement is criticized by the authors of the study?”
Some clarifications would be needed in Table 1:
· - Age group in years: Instead of age values, large numbers are presented. Should we rather read “Participants per age group”?
· - Please give the unit for ADL, ADL disabled and IADL or explain what is represented by these scores.
· - BMI: It would be relevant to indicate the unit (kg/m2) even if it is well known.
· - Change in protein intake: Please indicate the unit of the three variables documenting the change in protein intake at the end of the table.
Reviewer 2 Report
The paper analyzed the association of changes in protein intake and cognitive impairment among Chinese older adults. The research is well done in general and has a good potential to contribute to the field. I raise a few concerns for the authors to improve the manuscript.
First of all, the focus on older people should be clearly mentioned in the introduction. The current introduction did not clearly highlight the current study is specifically on old age. I also hope the authors could briefly mention about the necessity of distinguishing different sources of protein to better justify the following analyses. The first sentence of the third paragraph appears confusing. I guess the authors tried to point out the limitation of the current literature.
My major concerns are on the methodical side. Although the authors mentioned in the limitation statement that this study is not causal analyses. However, I think the authors should do their best in this analysis to show the effect of protein intake to cognitive functioning, not the other way around. In this regard, it is less clear in the method statement for some important method issues. Firstly, are time-varying variables accounted for in the analyses? Some of the dietary changes could be caused by the health and life events such as having a disease or losing a spouse. Trying to control for these issues could better reveal of the effect of changes in protein intake.
Second, it seems that the events of cognitive impairment could happen both in 2011 and 2014. This weakens the power of analyses, as the events may happen simultaneously as the protein intake changes (measured by the difference between 2008 and 2011). I will suggest the authors at least try a sensitivity analysis, directly modelling the change in protein intake in 2008 and 2011 and the cognitive status in 2014. The model could use MMSE scores as continuous variable with a fixed effect model. Such analyses could reinforce the evidence of causal effects. Third, the authors need to be more serious in deleting the cases of death and being lost to follow-up. Will this bias the results in the COX analyses? At least some discussions are needed because these cases are not missing randomly.
I also have a few minor issues as below. The definition of “extreme decline, moderate decline, mild decline, stable, mild improvement, moderate improvement, and extreme improvement” is missing. CLHLS should have centenarian samples, and please explain why this is dropped in this analysis. “Converter” in table 4 needs to be defined. More discussions are needed for the patterns of nuts and milk, which are not inline with the other patterns. The article needs careful proofreading and there are some obvious language mistakes such as line 305.
